# Usage of FTA^®^ Classic Cards for Safe Storage, Shipment, and Detection of Arboviruses

**DOI:** 10.3390/microorganisms10071445

**Published:** 2022-07-18

**Authors:** Janina Krambrich, Emelie Bringeland, Jenny C. Hesson, Tove Hoffman, Åke Lundkvist, Johanna F. Lindahl, Jiaxin Ling

**Affiliations:** 1Department of Medical Biochemistry and Microbiology, Zoonosis Science Center, Uppsala University, P.O. Box 582, 751 23 Uppsala, Sweden; emeliebringeland@gmail.com (E.B.); jenny.hesson@imbim.uu.se (J.C.H.); tove.hoffman@medsci.uu.se (T.H.); ake.lundkvist@imbim.uu.se (Å.L.); johanna.lindahl@imbim.uu.se (J.F.L.); jiaxin.ling@imbim.uu.se (J.L.); 2Department of Biosciences, International Livestock Research Institute, P.O. Box 30709, Nairobi 00100, Kenya; 3Department of Clinical Sciences, Swedish University of Agricultural Research, P.O. Box 7054, 750 07 Uppsala, Sweden

**Keywords:** biosafety, flavivirus, alphavirus, arboviruses, FTA^®^ Classic Card, RNA stability

## Abstract

Infections caused by arthropod-borne RNA viruses are overrepresented among emerging infectious diseases. Effective methods for collecting, storing, and transporting clinical or biological specimens are needed worldwide for disease surveillance. However, many tropical regions where these diseases are endemic lack analytical facilities and possibility of continuous cold chains, which presents challenges from both a biosafety and material preservation perspective. Whatman^®^ FTA^®^ Classic Cards may serve as an effective and safe option for transporting hazardous samples at room temperature, particularly for RNA viruses classified as biosafety level (BSL) 2 and 3 pathogens, from sampling sites to laboratories. In this study, we investigated the biosafety and perseverance of representative alpha- and flaviviruses stored on FTA^®^ cards. To evaluate the virus inactivation capacity of FTA^®^ cards, we used Sindbis virus (SINV), chikungunya virus (CHIKV), and Japanese encephalitis virus (JEV). We inoculated susceptible cells with dilution series of eluates from viral samples stored on the FTA^®^ cards and observed for cytopathic effect to evaluate the ability of the cards to inactivate viruses. All tested viruses were inactivated after storage on FTA^®^ cards. In addition, we quantified viral RNA of JEV, SINV, and tick-borne encephalitis virus (TBEV) stored on FTA^®^ cards at 4 °C, 25 °C, and 37 °C for 30 days using two reverse transcriptase quantitative PCR assays. Viral RNA of SINV stored on FTA^®^ cards was not reduced at either 4 °C or 25 °C over a 30-day period, but degraded rapidly at 37 °C. For JEV and TBEV, degradation was observed at all temperatures, with the most rapid degradation occurring at 37 °C. Therefore, the use of FTA^®^ cards provides a safe and effective workflow for the collection, storage, and analysis of BSL 2- and 3-virus RNA samples, but there is a risk of false negative results if the cards are stored at higher temperatures for long periods of time. Conscious usage of the cards can be useful in disease surveillance and research, especially in tropical areas where transportation and cold chains are problematic.

## 1. Introduction

Arboviruses, viruses transmitted by arthropods, are among the most important agents of tropical and emerging infections [1], and human, animal-derived, and vector samples containing arboviral material have to be transported from sampling sites to laboratories worldwide. The vast majority of emerging infections are caused by single-stranded RNA viruses [2], primarily due to their remarkable ability to generate divergent progeny within a short period of time and thus adapt to new environments [3]. Prominent arboviruses include several different flaviviruses, such as tick-borne encephalitis virus (TBEV), Japanese encephalitis virus (JEV), Zika virus (ZIKV), dengue virus (DENV), Usutu virus (USUV), and West Nile virus (WNV), as well as alphaviruses, such as Sindbis virus (SINV) and chikungunya virus (CHIKV). With economic globalization, the flourishing tourism industry, and climate change, the risk of spread of arboviruses and their vectors also to temperate countries has increased significantly [4,5,6,7]. Laboratory networks for improved surveillance and rapid identification of arbovirus infections are needed to provide early warning of the potential circulation and spread of disease in and to new areas [8]. The geographic spread of most of these viruses is constantly expanding, e.g., the increasing transmission of WNV in central Europe [9,10], the increasing geographic spread of CHIKV infections throughout the Americas [11], and the unprecedented emergence of ZIKV infections in the Pacific islands and Latin America [12].

Many arboviruses cause potentially severe human disease and therefore require handling with a high degree of biosafety [13,14]. In addition, RNA viruses are fragile and susceptible to degradation, especially in the absence of protective reagents and at high temperatures, which can result in the loss of high-quality material for analysis. In tropical regions, it is usually difficult to store and transport samples, such as cell culture supernatant, whole blood, serum, cerebrospinal fluid, semen, swabs, plant materials, insect homogenates, or various bacterial samples, in a continuous cold chain from the field to a biosafety laboratory [14], and it can be difficult to maintain desired biosafety standards. Collecting samples at sampling sites and shipping them to laboratories around the world for analysis presents another challenge, both from a biosafety perspective and in terms of preventing material loss. A safe and effective method for collecting and shipping arbovirus samples is therefore urgently needed. There have been innovations to circumvent this problem, such as the RNA stabilization solution RNA*later*, which can preserve and protect RNA in fresh tissue samples at room temperature, but unfortunately, some viruses remain infectious in RNA*later* [15,16]. Here, we assess another method for stabilizing viral RNA on Whatman^®^ FTA^®^ Classic Cards. This technology consists of filter paper impregnated with a patented chemical mixture of chemical denaturants and a free radical scavenger. The chemical mixture lyses cells and organelles, prevents bacterial overgrowth, and denatures proteins. The free radical scavenger helps protect cells and nucleic acids from the damage caused by free radicals. RNA is trapped in the matrix and remains tightly bound while proteins and inhibitors are washed out [17].

In this study, the stability of viral RNA and inactivation of viral infectivity on these FTA^®^ cards, using cell culture supernatant-derived virus samples, were studied under different time and temperature conditions, using reverse transcription quantitative polymerase chain reactions (RT-qPCRs) and cell viability assays. In the field, FTA^®^ cards could also be used for transporting various other samples. We used four arboviruses: two flaviviruses (JEV and TBEV) and two alphaviruses (SINV and CHIKV). These four arboviruses were selected because of their importance as causative agents of endemic vector-borne diseases. TBEV is transmitted by ticks (e.g., *Ixodes ricinus*) and can cause severe encephalitis in humans. The European subtype has a mortality rate of 1–2%, while the Far Eastern variant has a reported mortality rate of up to 30% [18,19,20]. JEV is an important cause of viral encephalitis in Asia, and *Culex tritaeniorhynchus* is one of the primary mosquito vectors, while pigs and birds are the main amplifying hosts [21]. Routine clinical diagnosis of these two infections is based on serological assays [22,23,24], but because of the close antigenic relationship between viruses such as TBEV and other flaviviruses, routine diagnosis is complicated due to serological cross-reactivity [25]. SINV is transmitted by mosquitoes (*Culex* species); genotype I, which was introduced to northern Europe from central Africa in the 1920s, can cause disease with symptoms such as arthralgia, rash, and fever [26,27]. CHIKV, which manifests itself with symptoms similar to SINV, is primarily transmitted to people through bites of infected *Aedes aegypti* and *Ae. albopictus* mosquitoes. CHIKV has caused outbreaks in Africa; the Americas; Asia; Europe; and the Caribbean, Indian, and Pacific Oceans [28].

## 2. Materials and Methods

### 2.1. Viruses

Two alphaviruses, CHIKV (Mal lineage) and SINV (strain 09-M-358, genotype I, Genbank accession number MK045245), and two flaviviruses, JEV (strain Nakayama, FD 007V-03215) and TBEV (strain 1993-3386) [18,27], were used to evaluate the biosafety and usability of Whatman^®^ FTA^®^ Classic cards (Figure 1). Viral stock titers were 1.15 × 10^6^ plaque forming units (PFU)/mL for SINV, 2.15  ×  10^5^ tissue culture infective dose 50 (TCID_50_)/mL for JEV, 1.0  ×  10^3^ TCID_50_/mL for TBEV, and 3.16  ×  10^7^ TCID_50_/mL for CHIKV. All SINV experiments were performed on Vero cells under BSL-2 conditions and all CHIKV, JEV, and TBEV experiments were performed on Vero E6 cells under BSL-3 laboratory conditions at the Zoonosis Science Centre (ZSC), Uppsala University, Sweden.

### 2.2. Whatman^®^ FTA^®^ Classic Cards

Whatman^®^ FTA^®^ Classic (hereafter FTA^®^) cards from GE Healthcare, Chalfont St Giles, UK, were used in this study. The chemical reagent on the cards is designed to lyse cells on contact, denature proteins, and protect nucleic acids from degradation. The FTA^®^ cards contain chemical denaturants and a free radical scavenger that tightly binds the nucleic acid while proteins and inhibitors are washed out of the matrix.

### 2.3. Biosafety Using FTA^®^ Cards

Infectivity tests were performed to test viral inactivation after drying on FTA^®^ cards. These tests were performed to distinguish between cell toxicity caused by the components of the FTA^®^ card itself and a potential cytopathic effect (CPE) caused by the virus inoculated onto the FTA^®^ card. A volume of 125 µL of SINV, JEV, and CHIKV virus stock was transferred to the designated circles (discs) on the FTA^®^ cards and dried for 2 h in a class II microbiological safety cabinet (MSCII). After two h of drying, one-eighth of the discs, each containing SINV, JEV, or CHIKV, and one FTA^®^ control (without added virus) were excised immediately, placed in 1 mL phosphate-buffered saline (PBS) as described before [29,30], and stored overnight at 4 °C. The next day, the disc supernatants were diluted in a ten-fold dilution series with Dulbecco’s Modified Eagle Medium (DMEM) (Gibco™, Paisley, UK) supplemented with 2% fetal bovine serum (FBS) (Gibco™, Paisley, UK), and 1% penicillin-streptomycin (Gibco™, Paisley, UK). An equal amount of each virus was diluted to 10^−7^ in a ten-fold series, to be used as a positive control.

Monolayers of Vero (green monkey kidney cells (ATCC^®^ CCL-81™)) and Vero E6 cells (green monkey kidney cells, clone E6 (ATCC^®^ CRL-1586™)) grown in 96-well plates with DMEM (Gibco™, Thermo Fisher Scientific, Waltham, MA, USA) supplemented with 2% FBS (Gibco™, Thermo Fisher Scientific, Waltham, MA, USA) were inoculated with 100 µL of the dilution series of infectious virus, virus filter paper eluate, and pure filter paper eluate, respectively, and incubated for 1 h at 37 °C. After the initial absorption period, the inoculum was removed and replaced with fresh cell culture medium, and then cells were incubated at 37 °C until analysis. Cells were visually assessed for signs of CPE after 48 and 96 h. In addition, 100 µL supernatants were collected at 48 and 96 h post-infection (hpi) for extraction and RT-qPCR. All experiments were performed in triplicate.

### 2.4. RNA Stability on FTA^®^ Cards

To evaluate the stability of viral RNA in the chemical matrix of the FTA^®^ cards, designated circles on the FTA^®^ cards were inoculated with 125 μL stock solution of two flaviviruses (JEV and TBEV) and one alphavirus (SINV), respectively. All cards were dried in an MSCII for 2 h. Cards were then sealed in plastic bags and stored at 4 °C, 25 °C, or 37 °C for 1, 7, or 30 days. The collected FTA^®^ cards were stored at −80 °C until further analysis.

### 2.5. Extraction of Viral RNA from FTA^®^ Cards

Before starting RNA extraction, we used SINV and evaluated the RNA extraction efficiency of the virus using different RNA extraction methods, including the QIAamp Viral RNA Mini Kit (Qiagen, Hilden, Germany) and the RNeasy Mini Kit (Qiagen, Hilden, Germany), TRIzol™ LS Reagent (Invitrogen, Thermo Fisher Scientific, Waltham, MA, USA), TRIzol™ Reagent (Invitrogen, Thermo Fisher Scientific, Waltham, MA, USA) and the Direct-zol™-96 RNA kit (Zymo Research, Irvine, CA USA), to see if they were compatible with the FTA^®^ cards. We found that the recovery efficiency of SINV viral RNA was best with the RNeasy Mini Kit. The QIAamp Viral RNA Mini Kit, the Direct-zol™-96 RNA kit, the TRIzol™ LS Reagent, and the TRIzol™ Reagent all had similar, slightly lower RNA recovery rates. However, after application to the FTA^®^ cards, the recovery efficiency of SINV viral RNA with the RNeasy Mini Kit decreased to a similar level to the other kits previously and remained at a similar level with the TRIzol™ LS Reagent, the TRIzol™ Reagent, and the Direct-zol™-96 RNA kit. Recovery efficiency after application to FTA^®^ cards was substantially lower with the QIAamp Viral RNA Mini Kit compared to all other methods.

RNA extraction of SINV and TBEV was performed using the RNeasy Mini Kit according to the manufacturer’s instructions. For TBEV, the experiment was performed in a BSL-3 laboratory before transferring samples inactivated in lysis buffer to a BSL-2 laboratory. The JEV samples were instead extracted using the Direct-zol™-96 RNA kit according to the manufacturer’s instructions, as inactivation of higher concentrations of virus with the RLT lysis buffer used by the RNeasy Mini Kit had been shown to be insufficient in some studies [31,32].

For both extraction methods, half of one inoculated disc was excised and stored in RLT lysis buffer (SINV and TBEV) or TRIzol™ LS Reagent (JEV), respectively, overnight at 4 °C before starting the extraction process. RNA was stored at −80 °C prior to analysis by RT-qPCR.

### 2.6. Reverse Transcription Quantitative PCR

Complementary DNA (cDNA) was synthesized using the RevertAid Reverse Transcriptase Kit (Thermo Fisher Scientific, Waltham, MA, USA) according to the manufacturer’s instructions. The 20 µL PCR mixture contained 4 µL 5X reaction buffer, 2 µL random primers, 2 µL 10 M dNTP, 1 µL RevertAid RT, 0.5 µL sterile water, and 10.5 µL extracted RNA. The cDNA was stored at −20 °C prior to qPCR analysis.

Portions of SINV nonstructural protein 1 (NS1) and JEV and TBEV NS5 protein were amplified using primers (Thermo Fisher Scientific, Waltham, MA, USA) previously described by Jöst et al., 2010 (SINV) and Patel et al., 2013 (JEV, TBEV) and the SsoAdvanced Universal SYBR^®^ Green Supermix Kit (BioRad, Irvine, CA, USA) [33,34]. The SINV forward primer was SIND F (5′-CAC WCC AAA TGA CCA TGC-3′) and the reverse primer SIND R (5′-KGT GCT CGG AAW ACA TTC-3′). The pan-flavi primers were Flavi all S (5′-TAC AAC ATG ATG GGG AAR AGA GAR AA-3′), Flavi all S2 (5′-TAC AAC ATG ATG GGM AAA CGY GAR AA-3′), and Flavi all AS4 (5′-GTG TCC CAG CCN GCK GTR TCR TC-3′). Reaction mixes contained 2 µL cDNA, 10 µL SsoAdvanced Universal SYBR^®^ Green Supermix (a buffer containing antibody-mediated hot-start Sso7d fusion polymerase, dNTPs, MgCI_2_, SYBR^®^ Green I dye, enhancers, stabilizers, and a blend of passive reference dyes (including ROX and fluorescein)), 1 μL forward and reverse primers, respectively (10 μM stock concentrations), and 6 μL nuclease-free water. The RT-qPCR assay was performed on a CFX96 Touch^TM^ Real-Time PCR Detection System (Bio-Rad Laboratories, Hercules, CA, USA) under the following conditions: initial activation at 95 °C for 3 min, followed by 45 cycles of denaturation at 95 °C for 5 s and annealing, extension, and fluorescence signal collection at 60 °C for 30 s. The run was completed with a melting curve analysis. Relative fluorescence unit (RFU) data were obtained using CFX Maestro^TM^ software (Bio-Rad CFX Maestro for Mac 1.1 version 4.1.2434.0214, Bio-Rad Laboratories, Hercules, CA, USA). Sanger sequencing at Macrogen (Amsterdam, the Netherlands) was used to sequence the amplicons. Sequencing results were analyzed using the Basic Local Alignment Search Tool (BLAST) at https://blast.ncbi.nlm.nih.gov/Blast.cgi (accessed on 1 January 2022). Cycle threshold (Ct) values were used as a measure for RNA quantity.

To quantify the copy numbers of the viral RNA of SINV, synthetic gBlock gene fragments (Integrated DNA Technologies, IDT, Coralville, IA, USA), corresponding to the primer binding region of SINV were constructed and used to generate a standard curve.

Tenfold dilution series from 1 × 10^9^ copies/μL to 1 × 10^1^ copies/μL were prepared in sterile water. Ct values for each dilution were measured and plotted against the logarithm of the copy numbers. Copy number, R-squared (R^2^), and amplification efficiency (E) values were generated by CFX Manager^TM^ software and used to evaluate RT-qPCR assays.

### 2.7. Data Visualization

All data were analyzed using R studio (R) (Rstudio, PBC, Boston, MA, USA). The packages ggplot2 and tidyverse were used for analyses and visualization [35,36,37]

## 3. Results

To study the inactivation of viral infectivity and the stability of viral RNA on FTA^®^ cards, the virus was inoculated onto the filter paper and subsequently eluted. The results were examined using cell viability assays and RT-qPCRs. To investigate the stability of viral RNA on the FTA^®^ cards, the virus was stored on the filter papers for up to one month at three different temperatures.

### 3.1. Cell Viability Assay to Determine the Inactivation of Samples Eluted from the FTA^®^ Cards

To determine the biosafety of virus samples transported and stored on FTA^®^ cards, the potential infectivity of the sample eluates was assessed. In a cell viability assay, the CPE caused by various dilutions of virus was compared to the cell death observed when eluates from virus samples stored on FTA^®^ cards were inoculated. To distinguish between CPE caused by the virus and CPE caused by the chemical mixture impregnating the FTA^®^ filter paper, an inoculation control was performed consisting of filter paper eluate only. Monolayers of Vero E6 cells were inoculated with dilution series of infectious virus, virus filter paper eluate, and filter paper eluate only and incubated at 37 °C. At inspection after 48 and 96 h, no difference in CPE development could be observed. Supernatant was collected at 48 and 96 h after inoculation, extracted, and subjected to RT-qPCR to verify the absence of replicating virus in the filter paper eluates. No replication (meaning no relative fluorescence and resulting Ct value) was detected in any FTA^®^-stored and eluted virus samples on cells 48 and 96 h post inoculation, whereas the virus controls were all PCR-positive with decreasing Ct values. Infectious CHIKV showed virus-induced CPE development down to a dilution of 10^−6^ and JEV down to a dilution of 10^−4^ (in two out of three replicates) (Table 1A). The CHIKV and JEV filter paper eluates showed CPE development down to a dilution of 10^−2^ (Table 1A). CPE caused by the chemical mixture on the control filter paper was also observed down to a dilution of 10^−2^ (Table 1A). The morphology of the observed CPE in the virus filter paper eluates was consistent with the morphology of the filter paper control eluate (Figure 2). The cells were elongated and detached from the bottom in flocs. In contrast, a typical virus-induced CPE was observed in the infectious virus samples, where the cells became smaller and round (Figure 2). The morphological similarity of the CPE observed in the virus filter paper and control filter paper eluates and the observation of CPE at much lower dilutions of the infectious viruses indicate that the chemical mixture on the FTA^®^ cards successfully inactivated the alpha- and flaviviruses tested.

In this study, we used JEV with a stock titer of 2.15 × 10^5^ TCID_50_/mL, and our in vitro infectivity experiment showed that FTA^®^ cards were able to inactivate all virus at least at a dilution of 10^−2^. This means that FTA^®^ inactivates the virus at least up to a titer of 2.15 × 10^3^ TCID_50_/mL, indicating that the inactivation efficiency of FTA^®^ cards for JEV in our setting was at least (1 − 1/(2.15 × 10^3^)) × 100% = 99.95%. For CHIKV, where a stock titer of 3.16 × 10^7^ TCID_50_/mL was used and inactivation was also observed at 10^−2^, the inactivation efficiency of FTA^®^ in our experiment was at least (1 − 1/(3.16 × 10^5^)) × 100% = 99.99%.

The same experiment was performed with SINV, and Vero cells exposed to SINV showed visible virus-induced CPE down to a dilution of 10^−4^. Chemically induced cell death was eliminated at a dilution of 10^−3^. The SINV filter paper eluate showed CPE development down to a dilution of 10^−2^, indicating that SINV was also inactivated upon contact with the chemical mixture on the FTA^®^ card (Table 1B). For SINV, the inactivation efficiency of FTA^®^ in our experiment was therefore at least (1 − 1/(1.15 × 10^3^)) × 100% = 99.99%.

### 3.2. Investigation of the Stability of Alpha- and Flavivirus Viral RNA on the FTA^®^ Cards

Stock solutions of SINV, TBEV, and JEV samples were each placed on FTA^®^ cards and stored at three different temperatures (4 °C, 25 °C, and 37 °C) for 1, 7, and 30 days, respectively. After the storage period, samples were eluted from the filter papers and RNA was extracted and analyzed using two different RT-qPCRs. Each of the stock solutions used to inoculate the cards was also extracted and used as baseline references. The Ct values of the RT -qPCR were used as a measure of RNA quantity. Based on the gBlock standard curves, the SINV RT-qPCR had a detection limit of 102 copies/µL.

All viruses showed a substantial increase in Ct value (i.e., decrease in RNA concentration) between stock solutions and the day one eluate of FTA^®^ cards at all temperatures (ΔCt_SINV_ 6.39, ΔCt_TBEV_ 7.14, and ΔCt_JEV_ 12.26) (Figure 3). In addition, a temperature-dependent degradation of RNA was observed by an increase in Ct value after one day for JEV and TBEV, with RNA being most stable at 4 °C and less stable at 25 °C and 37 °C. SINV RNA, on the other hand, degraded gradually over 30 days at 37 °C, but no substantial difference in RNA amounts was observed over 30 days for samples stored at 4 °C and 25 °C, respectively.

## 4. Discussion

This study investigated the safety and performance of storing arbovirus cell culture samples on FTA^®^ cards and found that the cards were safe to use for the flavivirus (JEV) and alphaviruses (CHIKV and SINV) tested. Varying RNA stabilizing performance at different temperatures over time was noted. Sample preservation and storage are necessary for conducting studies on RNA viruses, but this can be challenging, especially when conducting fieldwork in tropical regions. The FTA^®^ cards were developed using Whatman FTA^®^ technology and are used in a variety of settings, primarily for transporting various samples such as blood, tissue, food products, serum, and cell culture supernatant. The cards were developed for nucleic acid-related research, such as genetic identification, plasmid screening, and virus collection [38], and can have a wide range of applications in research. In many studies, FTA^®^ cards have been used to transport clinical and field samples for subsequent molecular detection or genotyping of various viruses, including enveloped and non-enveloped RNA viruses such as avian influenza virus (AIV), rotavirus A, measles virus, and rubella virus [30,39,40,41]. The cards have also been used for arbovirus observation, including Alfuy virus, Kunjin virus, Ross River virus, and Barmah Forest virus surveillance [42,43]. However, to our knowledge, no previous evaluations have been reported for SINV, CHIKV, TBEV, and JEV, which are all important alpha- and flaviviruses pathogenic to humans.

Safety is a major concern when transporting and handling pathogenic viruses. Previous studies, e.g., by Kraus et al. [17] and Rogers et al. [44], have shown that samples containing pathogens such as AIV and different strains of *Staphylococcus* and *Escherichia coli* bacteria are inactivated when placed on FTA^®^ cards. To date, only one of 14 studies on different viruses has reported that the examined virus, infectious bursal disease virus, was still infectious after storage on FTA^®^ cards [29,45]. One problem when using eluates from FTA^®^ cards to infect Vero or Vero E6 cells is that the eluate is toxic to the cells. To address this issue, we used different dilutions of the card eluates to show that viruses from FTA^®^ cards did not develop CPE beyond what is caused by the chemical mixture on the cards themselves and confirmed the absence of underlying replication by showing no Ct value in the RT-qPCR assay over time. Toxicity for Vero E6 cells and for Vero cells was completely eliminated at dilutions of 10^−2^ and 10^−3^, respectively. Therefore, we could not detect any CPE caused by infectious viral particles when the samples contained a virus titer of less than 10^4^ TCID_50_/mL. In addition, we collected the supernatant at two time points post-infection (48 h and 96 h) and were able to prove the absence of underlying replication in all the critical dilutions, in which potential virus-induced CPE might have been hidden behind the FTA^®^ toxicity.

According to the U.S. Environmental Protection Agency’s Disinfection Profiling and Benchmarking guidelines [46], a process must achieve 4-log (99.99%) removal and/or inactivation of viruses through removal (sedimentation and filtration) and/or inactivation (disinfection) to meet their standards. Given the 99.99% inactivation efficiency for SINV and CHIKV, the FTA^®^ cards achieve the required reduction. The slightly lower inactivation efficiency for JEV (99.95%) does not meet the required standards. However, with the JEV titers of 10^3^ TCID_50_/mL that we have, we cannot achieve the 4-log inactivation required to demonstrate complete inactivation. We consider the inactivation efficiency of all tested viruses as sufficient to conclude inactivation on the FTA^®^ cards.

One limitation of the present study is that repeated blind passages of the eluted samples for duplicate detection of inactivation was not performed. Blind passage is the gold standard for isolating viruses from, for example, suspected clinical samples, but given that the objective of this study was to evaluate the safety of FTA^®^ cards for shipment, storage, and detection of arboviruses, the lack of detection of virus in cell culture evidenced by both lack of CPE and no PCR positivity after 96 h indicates the cards should be safe for this purpose.

In this study, we used viruses from cell cultures, which is different from clinical samples because clinical samples often have different concentrations and more complex matrices such as protein-loaded body fluids or tissue samples. It is therefore advisable to analyze the effect of the matrix on the inactivation efficacy of the FTA^®^ cards before use. However, the viral titers used in this study were comparable to or higher than documented titers found in biological samples or experimental infection settings. The mean viral titer for CHIKV-infected humans was reported to be 5.6 × 10^5^ PFU/mL in a study conducted in Thailand [47]. In an experimental infection of hamsters as a model for JEV infection, viral titers of 10^4.2^ PFU/mL were found [48]. Dengue viremia, another important flavivirus infection, has been reported to be between 10^2^ to 10^9^ copies/mL in humans [49,50,51,52]. Different conditions such as humidity, temperature, virus strain, sample matrix, cells used, and virus concentrations could have an impact on the performance of FTA^®^ cards, as shown in other studies [53,54,55]. Therefore, further studies are needed to analyze the effects of different conditions on performance before applying them in field studies under varying conditions.

Viral nucleic acid diagnostics using fluorescence-based quantitative PCR has become the gold standard for quantification of viral load [56]. In this study, previously described primers for the detection of alpha- and flaviviruses were used [33,34]. Synthetic gene segments (gBlocks) were additionally used to provide a standard curve for evaluation of the SINV RT-qPCR assay and to assess how many copies of viral RNA could be rescued from the FTA cards after storage. Based on the gBlock standard curve, the SINV RT-qPCR had a detection limit of 10^2^ copies/µL. By comparing the Ct values of the standard curve with the Ct values of the recovered samples, we believe that sufficient amounts of SINV RNA, between 10^4^ and 10^5^ copies of virus in the eluates, could be recovered from the FTA^®^ cards to be successfully used in downstream nucleic acid diagnostics such as whole genome sequencing. The quality (i.e., the integrity of RNA) would however also have to be examined for whole genome sequencing. We demonstrated that RNA from the viruses studied could be detected up to 30 days after application to the FTA^®^ card and that the best preservation occurred at lower temperatures. However, the stability of the RNA of SINV, JEV, and TBEV on the FTA^®^ card proved to be different. This could be due to the differences in genome structure between alphaviruses and flaviviruses, with alphaviruses in comparison to flaviviruses having a polyadenylated (poly-A) tail that potentially could protect the RNA from degradation [57]. The observed difference between the two flaviviruses cannot be explained within the scope of this study. These are two similar viruses treated in the same way, under the same conditions, and in the same experimental setup, but the results show a substantial difference in RNA stability over time. Therefore, we strongly advise other scientists to evaluate the performance of their specific viral target before using the FTA^®^ cards. There is a risk of losing material and obtaining a false negative result. The viral RNA of all flaviviruses is protected by the capsid (C) protein, but the homology of the JEV C protein with the C proteins of WNV, DENV, and TBEV is only 67%, 33%, and 25%, respectively [58]. This could be one factor leading to differences in degradation of the flavivirus RNA.

Storage at 4 °C is perhaps the safest recommendation for subsequent RNA analyses, but storage at 25 °C also gave good results for both SINV and TBEV.

## 5. Conclusions

Monitoring the spread of flavi- and alphaviruses, such as those used in this study, is very important for public health surveillance, and we here show how handling and transport can be made easier, cheaper, and safer. Arboviral inactivation and RNA stability when stored on FTA^®^ cards was studied over time and under different temperature conditions, using cell viability assays and RT-qPCR with cell culture supernatant. Our study demonstrates that FTA^®^ cards safely and efficiently can be used for the collection, storage, transport, and analysis of the examined arboviruses, which may facilitate future transport of biological samples. Since RNA stability was best at 4 °C, caution should be taken when transporting samples in higher temperatures, and the loss of potential quantity and quality of material should be considered. Different strains, concentrations, matrices, and environmental conditions may affect the performance of FTA^®^ cards. Therefore, it is recommended to thoroughly test the FTA^®^ cards with the desired matrix and analyze different conditions before use.

## Figures and Tables

**Figure 1 microorganisms-10-01445-f001:**
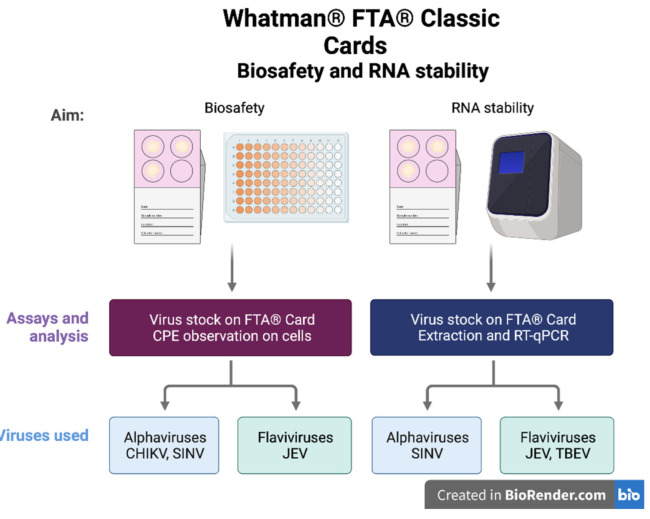
Overview of workflow. For biosafety, virus stocks were inoculated onto FTA^®^ cards and eluted, and then eluates were placed on cell monolayers to observe development of cell cytopathic effect. RNA stability was tested by inoculating stocks onto FTA^®^ cards, which were stored at different temperatures for different periods of time. RNA was then extracted and quantified in a reverse transcriptase quantitative PCR assay. Not all viruses were used in all assays, but representatives of each virus family were selected based on biosafety and availability. Workflow figure created in BioRender.com (accessed on 19 May 2022).

**Figure 2 microorganisms-10-01445-f002:**
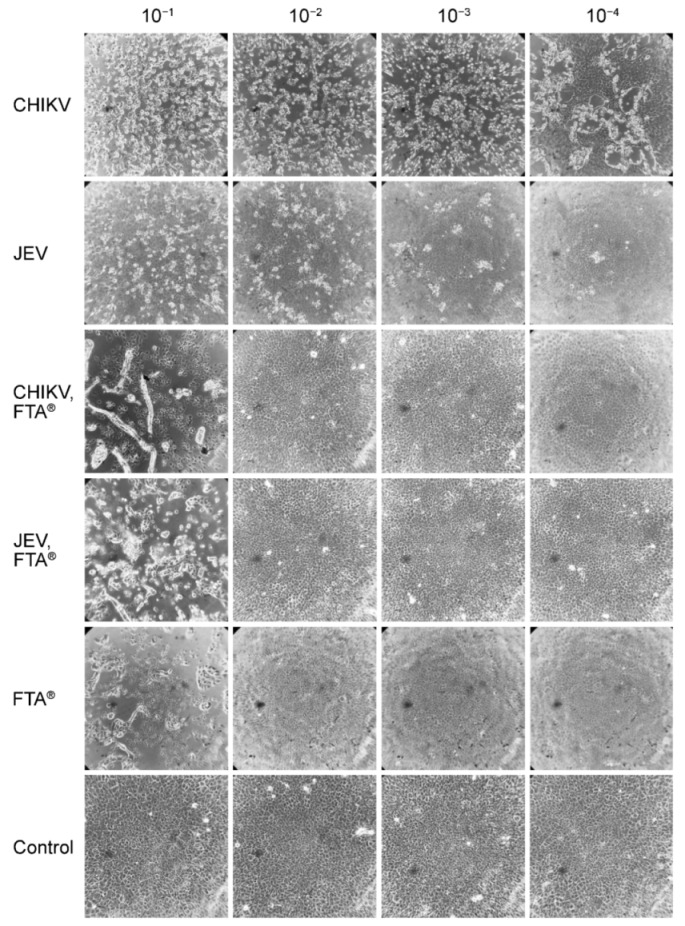
Cell cytopathic effect (CPE) development in Vero E6 cells infected with infectious and FTA^®^ inactivated Japanese encephalitis virus (JEV) or chikungunya virus (CHIKV). Virus was inoculated at various concentrations on Vero E6 monolayers and incubated for 48 h. Virus-induced CPE could be observed down to a dilution of 10^−4^. Chemically induced CPE could be observed in the FTA^®^ control down to a concentration of 10^−1^. The inactivated virus samples (JEV,FTA^®^ and CHIKV,FTA^®^) showed CPE development identical to the CPE in the FTA^®^ control. The supernatant was additionally collected and extracted to verify whether virus replication underlay the inactivated virus samples. No replication was detected. The wells shown are representative of all replicates, and the total magnification used to observe the cells was 100×.

**Figure 3 microorganisms-10-01445-f003:**
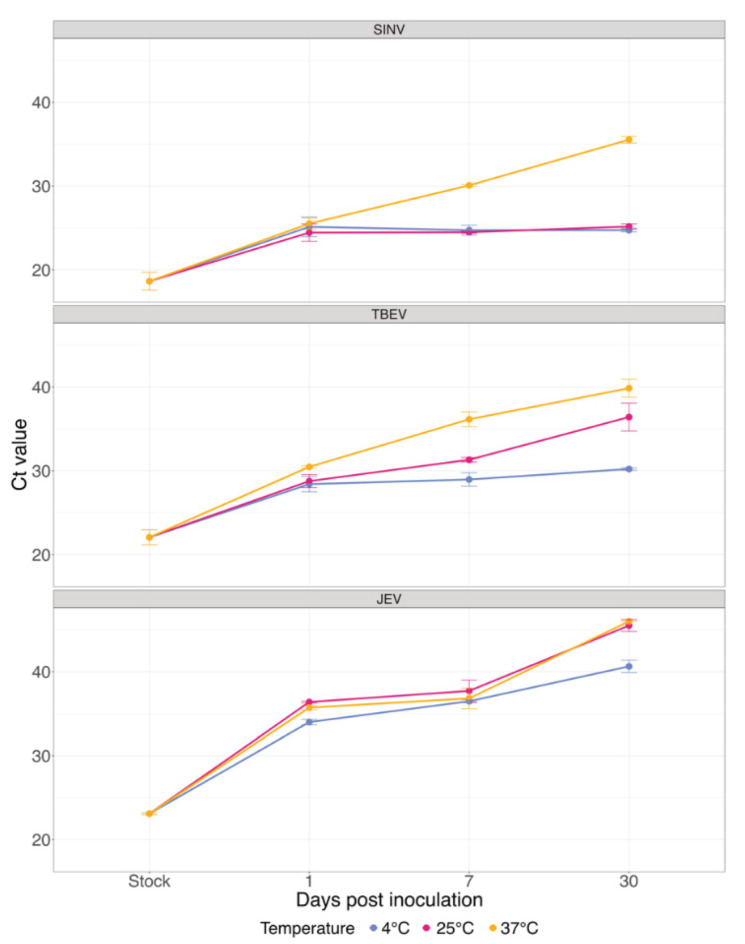
Temporal and temperature-based degradation of Sindbis virus (SINV), tickborne encephalitis virus (TBEV), and Japanese encephalitis virus (JEV) RNA stored on FTA^®^ cards, visualized by the change in cycle threshold (Ct) value over time. Points are average Ct values of all replicates and the error bars are the standard deviation.

**Table 1 microorganisms-10-01445-t001:** Documentation of observed cell cytopathic effects (CPE) after 48 and 96 h in (**A**) Vero E6 cells caused by Japanese encephalitis virus (JEV) or chikungunya virus (CHIKV) and (**B**) Vero cells caused by Sindbis virus (SINV) compared to CPE observed in cells incubated with viruses from FTA^®^ card eluates and FTA^®^ card eluates only. CPE caused by the chemical mixture is indicated by exclamation marks, plus signs indicate the observation of virus induced CPE, and minus signs mean that no CPE was observed and the cells look like the cell control. Triplicates were analyzed.

(**A**) Vero E6 cells															
	**FTA^®^**	**FTA^®^**	**FTA^®^**	**FTA^®^** **JEV**	**FTA^®^** **JEV**	**FTA^®^** **JEV**	**FTA^®^** **CHIKV**	**FTA^®^** **CHIKV**	**FTA^®^** **CHIKV**	**JEV**	**JEV**	**JEV**	**CHIKV**	**CHIKV**	**CHIKV**
1	!	!	!	!	!	!	!	!	!	+	+	+	+	+	+
10^−1^	!	!	!	!	!	!	!	!	!	+	+	+	+	+	+
10^−2^	−	−	−	−	−	−	−	−	−	+	+	+	+	+	+
10^−3^	−	−	−	−	−	−	−	−	−	+	+	+	+	+	+
10^−4^	−	−	−	−	−	−	−	−	−	+	+	−	+	+	+
10^−5^	−	−	−	−	−	−	−	−	−	−	−	−	+	+	+
10^−6^	−	−	−	−	−	−	−	−	−	−	−	−	+	+	+
10^−7^	−	−	−	−	−	−	−	−	−	−	−	−	−	−	−
(**B**) Vero cells															
	**FTA^®^**		**FTA^®^**		**FTA^®^**	**FTA^®^** **SINV**	**FTA^®^** **SINV**		**FTA^®^** **SINV**		**SINV**		**SINV**		**SINV**
1	!		!		!	!	!		!		+		+		+
10^−1^	!		!		!	!	!		!		+		+		+
10^−2^	!		!		!	!	!		!		+		+		+
10^−3^	−		−		−	−	−		−		+		+		+
10^−4^	−		−		−	−	−		−		+		+		+
10^−5^	−		−		−	−	−		−		−		−		−
10^−6^	−		−		−	−	−		−		−		−		−
10^−7^	−		−		−	−	−		−		−		−		−

## Data Availability

Data will be made available from the corresponding author upon request.

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
