# Peer review of "Usage of FTA^®^ Classic Cards for Safe Storage, Shipment, and Detection of Arboviruses"

_microorganisms, 2022, doi:10.3390/microorganisms10071445_

Round 1
Reviewer 1 Report
see the attachment

Author Response
We thank you for reviewing the manuscript and for the good feedback on our study. We do agree that future projects that wants to use these cards indeed need to evaluate them for the matrices in questions and under different conditions to optimize the process for each virus-matrix combination. We have updated our conclusions to make these recommendations clear. We have responded to and answered the highlighted points in the attached document.
Reviewer 2 Report
This is a well written manuscript, confirming the suitability of FTA cards for storage and transport of samples for testing for a range of viruses, and highlights that parameters should be assessed for individual viruses, even within the same group (eg. flaviviruses).
The following two points should be however, discussed in the context of this work:
1) For biosafety purposes, inoculated cards should be tested for viral inactivation as soon as possible after inoculation/drying to assess the safety of handling cards. See comment Line 125. Also, inactivation assays are traditionally tested over 3 passages in appropriate cell lines.
2) Humidity was not considered. Manufacturers recommend the use of special envelopes and/or desiccant when storing cards at RT. This might have improved stability of viral RNA on cards kept at room temperature in this study.
Specific points:
Line 55 …cause potentially severe…
Lines 84-86 What is the relevance of serology in this context?
Line 99 Create a separate paragraph for FTA cards since they are the feature of this study and should not be buried under the information on viruses.
Lines 100-102 - PFU/mL versus TCID50/mL – why were different assays used?
Line 114 Was storing cards in PBS at 4°C overnight the whole elution process? Needs to be referenced.
Line 120 State volume used for this experiment. 96 well plates?
Something to think about: Applying the 125 uL of virus suspension onto one FTA card and then using 8 segments of that card could eschew results. Depending on how the liquid was pipetted onto card, the distribution of viruses might not be even within the circle. Quantifying virus load as a preliminary experiment on the eight parts would have shown if all portions were equal or not.
Line 125 Testing earlier time points for inactivation are necessary rather than waiting for 2 and 4 days. You want to know the earliest timepoint the cards are safe to handle: inactivation after the drying period of 2-3 hours is most desirable since the cards can then safely be handled or sent away.
Line 132 Plastic bags are not ideal for storage at higher temperatures as they can cause condensation, especially if there is still some moisture on cards. Desiccant should have been added as recommended by manufacturers.
Line 133 I understand that samples were frozen between inoculation and further processing. However, to show that RNA did not degrade after freeze/thawing, titres of samples should have been tested before freezing and compared with frozen/thawed samples.
Line 136 Why were two different extraction methods and two different lysis buffers used? There is no mention in the results section regards performance of the different methods or in discussion.
Line 247-253 Statistical analyses are needed to confirm ‘significant increase’ (L 247) and ‘no significant difference’ (L 252). For JEV, 37°C performs better than 25°C (D 1 and 7) and on D 30, Ct score for 37°C is just a fraction higher than 25°C, so ‘least stable at 37°C is inaccurate.
Table 1 Legend: Mention time frames tested
Author Response
We thank you for reviewing the manuscript and for the good feedback on our study. We have responded to and answered the highlighted points in the attached document, and added discussions of the highlighted aspects to the manuscript.
Round 2
Reviewer 1 Report
I thank the authors for submission of the improved version of the manuscript. In general, I can accept the adapted version for publication based on the clarifications in the text. Nevertheless, many points of a comprehensive method validation remain open and should be considered in future work.
Minor comments:
Line 147: Please use the correct name of the kits, e.g. QIAamp Viral RNA Mini Kit (Qiagen)
Line 151 and 152: What is the result of the comparative testing for the QIAamp Viral RNA Mini kit? This kit is missing here.
Line 155: Remove “Qiagen.”
Line 160/161: References for the statement should be added
Author Response
We thank you for getting back to us so quickly and for accepting the improved version of the manuscript for publication with minor changes. We will consider the good arguments on method validation in our future work. The point-by-point responses to the requested changes and edits are provided in the attached document.